# Omics-Driven Strategies for Developing Saline-Smart Lentils: A Comprehensive Review

**DOI:** 10.3390/ijms252111360

**Published:** 2024-10-22

**Authors:** Fawad Ali, Yiren Zhao, Arif Ali, Muhammad Waseem, Mian A. R. Arif, Obaid Ullah Shah, Li Liao, Zhiyong Wang

**Affiliations:** 1School of Breeding and Multiplication (Sanya Institute of Breeding and Multiplication), School of Tropical Agriculture and Forestry, Hainan University, Sanya 572025, China; fawadali365@gmail.com (F.A.); 18737449313@163.com (Y.Z.); m.waseem.botanist@gmail.com (M.W.); obaidus890@gmail.com (O.U.S.); 2Department of Plant Sciences, Quaid-I-Azam University, Islamabad 45320, Pakistan; arifali@bs.qau.edu.pk; 3Nuclear Institute for Agriculture and Biology College, Pakistan Institute of Engineering and Applied Sciences (NIAB-C, PIEAS), Jhang Road, Faisalabad 38000, Pakistan; m.a.rehman.arif@gmail.com

**Keywords:** abiotic stress, omics, climate change, genetic diversity, machine learning, speed breeding, food security

## Abstract

A number of consequences of climate change, notably salinity, put global food security at risk by impacting the development and production of lentils. Salinity-induced stress alters lentil genetics, resulting in severe developmental issues and eventual phenotypic damage. Lentils have evolved sophisticated signaling networks to combat salinity stress. Lentil genomics and transcriptomics have discovered key genes and pathways that play an important role in mitigating salinity stress. The development of saline-smart cultivars can be further revolutionized by implementing proteomics, metabolomics, miRNAomics, epigenomics, phenomics, ionomics, machine learning, and speed breeding approaches. All these cutting-edge approaches represent a viable path toward creating saline-tolerant lentil cultivars that can withstand climate change and meet the growing demand for high-quality food worldwide. The review emphasizes the gaps that must be filled for future food security in a changing climate while also highlighting the significant discoveries and insights made possible by omics and other state-of-the-art biotechnological techniques.

## 1. Introduction

A major abiotic stressor that negatively affects both agricultural productivity and global food security is an excessive concentration of salt. Soil salinity ranks among the top 10 risks to the world’s land resources, growing by 1.5 million hectares every year. Salinity stress is a global problem that jeopardizes productivity, food security, and ecosystems. Salinity affects several aspects of plant life, including physiology, molecular mechanisms, nutrient and water uptake, and seed germination. Anthropogenic activities and increasing industrialization worsen the lack of arable land, pushing agricultural production into in-appropriate conditions like salt-prone zones [1].

After drought, soil salinity is regarded as the second most significant abiotic stress. In lentils, salt exposure often causes a 20% to 100% reduction in plant development and seed production [2,3]. Salt stress inhibits plant growth and increases osmotic pressure, hindering the absorption of soil water by causing Cl-ion-specific toxicity and creating an imbalance of ions (Ca^2+^, K^+^, and Na^+^). This considerably lowers the yield of crops and causes serious discomfort to the morpho-physiological and biochemical traits of the plant.

Improving lentil productivity in the face of changing climatic conditions requires an understanding of how lentils respond to, adjust to, and endure salinity changes [4]. Novel ideas for breeding saline-smart lentils can be obtained by examining how lentils have evolved to withstand stress and developed survival strategies to mitigate the adverse impacts of salinity stress. Plants have an innate ability to control their own growth and development in response to both internal and external stimuli [5]. Many biotechnological approaches have been used in recent years to gain an understanding of the pathways and mechanisms underlying plant tolerance and responses to salt stress. Omics approaches (e.g., transcriptomics, proteomics, metabolomics, miRNAomics, phenomics, and ionomics) at the tissue or single-cell levels have become state-of-the-art techniques that could promote the improvement of lentils and help ensure global food security [6] (Figure 1). These cutting-edge omics techniques are essential to saline-smart breeding initiatives because they provide new understandings of the genetic, molecular, and physiological factors influencing plant responses to salt stress. These methods assist in developing saline-smart lentils with improved tolerance and yield, supporting sustainable agriculture, and addressing issues related to global food security by deciphering intricate molecular processes [7].

Similarly, the application of machine learning techniques [8,9,10] and speed breeding techniques has become popular in order to accelerate breeding programs and create new cultivars in a shorter time [11,12,13]. Machine learning is particularly useful for analyzing large-scale datasets and predicting intricate traits, which can lead to improved cultivar production and the ability to make well-informed decisions regarding the progression of traits, such as in the case of saline-smart lentils [8,9,10]. Speed breeding, on the other hand, shortens breeding cycles and speeds up plant growth, making it easier to quickly produce new cultivars. Additionally, speed breeding greatly accelerates conventional breeding timelines and enhances the rapid development of high-yield, stress-tolerant cultivars [11,12].

This review also discusses the impacts of salinity and climate change on lentil production. It outlines recent advancements in several omics strategies that may enhance the breeding and production of lentils under salt stress. Additionally, it investigates how machine learning, speed breeding, and single-cell omics-assisted breeding could hasten the development of saline-smart lentils. Finally, the review provides an updated overview of advanced breeding techniques that might be a game-changer for developing saline-smart lentil cultivars in order to fulfill future food demand.

The information compiled for this review is drawn from a diverse range of sources, including peer-reviewed scientific articles and technical reports from reputable organizations, accessible through scientific databases such as ResearchGate, Web of Science, Google Scholar, and PubMed. A comprehensive literature search was conducted in these databases using various search terms, including salt stress, global salinity, saline–alkaline stress, climate change, foliar uptake, stress, tolerance, avoidance, antioxidants, phytohormones, molecular mechanisms, omics, state-of-the-art techniques, legumes, lentils, genomics, transcriptomics, proteomics, metabolomics, miRNAomics, epigenomics, phenomics, ionomics, machine learning, and speed breeding. Most of the scientific articles included in this review were published between 2010 and 2024, except a few earlier studies, which were selected due to their unique scientific significance.

### Effect of Climate Change and Salinity Stress on Lentil Production

Climate change is an ongoing and significant issue that puts considerable pressure on various areas of the world economy. The agricultural industry is particularly vulnerable due to its heavy reliance on a stable climate, which could lead to significant global crop losses [14,15]. Salinity, temperature, drought, and flooding are examples of abiotic stresses that are becoming more frequent and severe due to climate change [14,15,16,17]. It is expected that these climate change consequences will have an impact on agricultural yields in the coming years.

Soil salinity affects 20% of all cultivated land and 33% of irrigated agricultural land globally. This has resulted in a significant decline in crop productivity in the salt-affected areas [18,19]. Furthermore, it is anticipated that by 2050, half of all cultivable land will be affected by salinity, as 10% of arable land is impacted by salinity annually due to both natural and man-made causes (Figure 2) [20]. Since the demand for food is expected to rise by 70% to 110% to accommodate the world’s growing population by 2050, food security has emerged as the most pressing challenge in agriculture [21]. This is because the rate of crop production may decline due to the development of saline areas [22,23]. Plant growth is impacted by salt stress, which initially causes osmotic stress, followed by ionic toxicity [24]. Leaf growth is reduced as salinity-induced osmotic stress inhibits cell division and expansion. Subsequently, mature leaves undergo chlorosis, necrosis, and senescence due to ionic toxicity [25]. Ion homeostasis is disturbed by extra Na^+^, which can compete with K^+^ for crucial substrate binding sites in enzymes, further impairing protein synthesis and enzyme performance [25,26]. Plants decrease water loss through stomatal conductance in response to osmotic stress, which causes a lack of CO_2_ availability that is necessary for the Calvin Cycle to function. Because of this, more light is absorbed than is needed for photosynthesis and photorespiration, which increases the generation of reactive oxygen species (ROS). The accumulation of toxic levels of Na^+^ and Cl^-^ also hinders the repair of photosystem II damage caused by excessive light, which results in the generation of ROS [27]. Excess ROS can lead to oxidative damage to proteins, lipids, and DNA; in extreme cases, it can even result in cell death [28]. Plants’ antioxidant defense mechanism, on the other hand, regulates the production of excessive ROS and keeps cells from harming their essential components. The combined action of enzymes like catalase (CAT), ascorbate peroxidase (APX), glutathione reductase (GR), monodehydroascorbate reductase (MDHAR), dehydroascorbate reductase (DHAR), and non-enzymatic antioxidants like ascorbic acid (AsA), glutathione (GSH), phenolic compounds, alkaloids, and α-tocopherols work in tandem to keep ROS levels below the point that results in cellular damage [29,30]. Upregulation of the antioxidant system may confer tolerance to plants under conditions of salt stress through the preservation of cellular redox equilibrium [29,31,32].

Understanding the distinctions between the damage caused by the osmotic and ionic components of salt stress is crucial to fully comprehend the mechanisms underlying salt stress. Growth reduction is a quick response to osmotic stress that frequently happens in a matter of minutes. In a few days, ion-specific damage to leaves manifests as necrosis, chlorosis, and senescence. This is because the excess Na^+^ accumulates in the cytosol of the leaf at a faster rate than the cell compartmentalizes to lessen ion toxicity [33]. Salt stress in lentils results in decreased plant total biomass, plant height, grain yield, anthocyanin coloration in stems and leaves, and a decrease in flowering and pod setting [34]. There is a need for a sustainable solution to salt stress in order to minimize lentil yield losses. An integrative omics-driven approach could be a possible solution to overcome salt stress.

## 2. Omics-Driven Breeding for Saline-Smart Lentils

The plant phenotype is influenced by the regulation of genes, proteins, metabolites, miRNAs, and epigenetic markers in response to salt stress. It has been possible to generate saline-resilient lentils more swiftly by utilizing integrated omics techniques, which have been widely investigated to clarify stress responses and tolerance mechanisms (Figure 1). In order to improve our understanding of genotype-environment interactions, or more accurately, gene-phenotype interactions, it is imperative to investigate alternative omics approaches, such as metabolomics, proteomics, phenomics, miRNAomics, epigenomics, and ionomics profiling, despite the remarkable advancements in genomics and transcriptomics. To ascertain the link between saline-stressed and stress-tolerant cultivars, it is imperative to explore various germplasm prior to undertaking large-scale omics integration. This information will be useful in determining which germplasm shows tolerance when exposed to salt stress. We have thoroughly evaluated the literature in the parts that follow in order to investigate the mechanisms underlying salt stress tolerance in lentils.

### 2.1. Lentil Genomics

Genomics is regarded as a classical omics approach that offers detailed information on the complete genome, as opposed to focusing on a particular gene or its product [35,36]. Complete genetic material information is the focus of genomics, which provides crucial information on three pivotal interactions (epistasis, pleiotropy, and heterosis). These data assist in identifying a set of genes that can be applied to agricultural enhancement programs [36,37,38,39].

#### 2.1.1. Molecular Markers, Genetic Diversity, and Population Structure

The genus *Lens* has yielded valuable insights into genetic diversity and evolutionary relationships through the use of multiple molecular marker systems. These systems encompass amplified fragment length polymorphisms (AFLPs), random amplified polymorphic DNAs (RAPDs), simple sequence repeats (SSRs), inter-simple sequence repeats (ISSRs), and restriction fragment length polymorphisms (RFLPs) [40,41,42,43]. Single nucleotide polymorphism (SNP) markers in lentils have been found using technologies such as Illumina Genome Analyzer, Illumina Golden Gate sequencing [44], and competitive allele-specific PCR [45]. Despite the introduction of various molecular markers, only a limited number have been studied for their role in investigating salt stress in lentils. Dissanayake et al. (2020) [46] used SNP markers, while two different investigations [47,48] utilized SSR markers to examine the effects of salt stress in lentils. The most significant factors that influence crop breeding effectiveness are genetic diversity, selection, and heritability. Utilizing the natural agro-biodiversity preserved in gene banks assists in increasing crop diversity. These collections are essential resources for finding beneficial genes and/or alleles, which are the foundation of any pre-breeding initiative. The International Center for Agricultural Research in the Dry Areas (ICARDA) genebank holds the largest collection of lentils in the world, conserving 14,597 *Lens* accessions, including 612 wild accessions, 11,405 landraces, and 2580 breeding lines (www.genebanks.org/genebanks/icarda) (accessed on 26 August 2024). The National Bureau of Plant Genetic Resources, India (NBPGR) (7712 acc., www.nbpgr.ernet.in) (accessed on 26 August 2024), the Australian Grains Genebank (AGG) (5254 acc., https://grdc.com.au) (accessed on 26 August 2024), The European Cooperative Programme for Plant Genetic Resources (ECPGR) (4598 acc., https://www.ecpgr.org/) (accessed on 26 August 2024), the United States Department of Agriculture (USDA) (3187 acc., www.ars.usda.gov), the Vavilov Institute of Plant Genetic Resources, Russia (VIR) (2556 acc., www.vir.nw.ru) (accessed on 26 August 2024), and Plant Gene Resources of Canada (PGRC) (1150 acc., https://pgrc.agr.cg.ca) (accessed on 26 August 2024) are organizations that also collect and preserve lentil germplasm. However, major gaps still exist in the germplasm at the species and genotype levels [49] (Figure 3). To address these gaps, a broader spectrum of underrepresented areas of diversity must be incorporated into the wild gene pool.

The investigation of genetic diversity and relationships between conserved germplasm has important impacts on identifying material that could be valuable to particular breeding requirements, especially in cultivated lentils with a narrow genetic base, as well as for enabling precise documentation of genetic resources. A significant amount of genetic variation for agro-morphological and phenological traits has been described in the *Lens* genetic resources [50,51]. However, a limited number of studies identified salt-tolerant accessions in lentils. To date, less than 100 lentil accessions have been recognized as salt-tolerant, as reported in various research works [47,48,50,52,53,54,55,56,57,58,59,60,61,62,63,64] (Figure 3) (Table 1). Several salt-tolerant genotypes have been identified across various studies. For instance, PSL-9 has been recognized as salt-tolerant by both Singh et al. (2017) and Singh et al. (2020) [47,48]. Similarly, PDL-1 was reported to exhibit salt tolerance by the same authors in both years, as well as Singh et al. (2021) [53]. The genotype Firat87 was noted for its salt tolerance by Kökten et al. (2010) and Seferoglu et al. (2013) [52,54]. Furthermore, Ustica and Pantelleria were identified as salt-tolerant by Sidari et al. (2007), while Panuccio et al. (2022) expanded this list to include Pantelleria, Ustica, and Castelnuccio di Norcia [63,64]. In contrast, Eston was classified as salt-sensitive by de Abreu et al. (2004) and Muscolo et al. (2015) [57,60]. Additionally, unique salt-tolerant genotypes have been reported in the research conducted by Dissanayake et al. (2021), Sinha et al. (2019), Sarker et al. (2003), Köse (2012), Gaafar and Seyam (2018), Al-Quraan and Al-Omari (2017), and Hossain et al. (2017) [50,55,56,58,59,61,62]. These findings highlight the genetic diversity and potential for breeding programs aimed at improving salt tolerance in lentils.

Several molecular marker systems have been used to study the genetic diversity of the genus Lens, but most of these studies were performed on small-scale germplasm and focused mostly on cultivated lentils. Alo et al. (2011) conducted one of the first extensive genomic investigations on both wild and cultivated lentils [65]. They provided evidence that domestication reduced genetic diversity by about 40% and supported the theory that southern Turkey is the origin of cultivated lentils. Khazaei et al. (2016) tested accessions from 54 countries and identified three main groups of cultivated lentils that represented the agro-ecological zones of the world: the northern temperate, Mediterranean, and South Asia (sub-tropical savannah) [66]. In addition, Pavan et al. (2019) found that phenotypic features, such as seed size and early flowering, in Mediterranean germplasm were correlated with geographic location and genetic clustering [67]. The findings of Khazaei et al. (2016) and Pavan et al. (2019) suggest that lentil diversity has been significantly influenced by both artificial and natural selection [67,68]. Dissanayake et al. (2020) found only a weak correlation between geographic origin and genetic relationships, in contrast to previous studies that confirmed the phylogenetic relationships of *Lens* species [42]. Genetic diversity among 467 wild and cultivated lentil accessions was assessed using 420,000 SNPs from gene transcripts. Morphological and phenological characteristics of over 76% of ICARDA lentil accessions have been characterized (www.genesys-pgr.org; [68,69,70,71]) (accessed on 26 August 2024). A lentil diversity panel (LDP) of 324 accessions was phenotyped for phenological features over two seasons in nine distinct locations worldwide as part of the Canadian-led AGILE project [72]. Additionally, an exome capture array was used for genotyping this collection [73], and seeds from 321 lines were deposited in the ICARDA genebank. This well-characterized set will be helpful for future research on trait variety, as will the mini-core and FIGS sets from ICARDA and the INCREASE collections. Studies at the geographic centers of origin, where there is more diversity, should be carried out to comprehend crop diversification and adaptation [74].

#### 2.1.2. Genomic Analysis of Lentil Under Salt Stress

Genetic progress in lentils is slow compared to other legume crops, including soybean, common bean, pigeon pea, and chickpea. Therefore, existing lentil breeding programs face challenges in fully implementing marker-assisted selection (MAS) due to the insufficient availability of genetic resources [75]. The primary barriers to genomics-enabled improvement in lentils include the large genome size, restricted genetic base, lack of candidate genes, low-density linkage map, and difficulties in identifying advantageous alleles.

Next-generation sequencing (NGS) technology has recently advanced, making it easier to create array-based high-throughput genotyping platforms incorporating SNP markers. A significant amount of NGS has been performed on the lentil cultivar CDC Redberry [76]. Scaffolds covering more than half of the genome (2.7 Gb of the anticipated 4.3 Gb) were created from an original draft of 23× coverage. Later, this was supplemented with 125× coverage. The initial 23× draft assembly was used to identify gene sequences for a number of phenotypes, and the generated SNP markers are now available for MAS in the lentil breeding programs [76]. Additionally, the tight evolutionary relationships with model legumes like *Lotus japonicus* and *Medicago truncatula* have discovered probable orthologous gene sequence resources in the lentil genome and provide a significant opportunity for comparative genome mapping [77,78]. Genomic tools and technologies have created new opportunities for the application of genomics-assisted selection in lentil breeding. Utilizing reverse genetic techniques presents an enormous opportunity for the development of lentil cultivars.

Genetic mapping and its introgression into genomics-assisted breeding processes help in developing saline-smart cultivars. Quantitative trait loci (QTL) mapping is a crucial approach to understanding the genetic basis of various complex phenotypes influenced by multiple genes. Because several genes influence tolerance to salt stress, it displays quantitative inheritance [79,80]. Genome-wide association mapping (GWAS) is another useful technique for reducing limitations. It offers the advantages of the diversity panel’s history of meiosis while potentially improving precision [81]. Association mapping is more practical and economical than bi-parental mapping [82]. Expanding knowledge of the genetic foundation of traits is crucial for enhancing crop salinity resistance [83].

Consequently, using direct selection for higher yield as a measure of salt resistance is inappropriate [84]. Genomics has become a cutting-edge technique for finding and selecting high-precision superior alleles and applying them in breeding programs [85]. Advances in NGS technology have accelerated the creation of lentil genomic resources between 2010 and 2020 [86,87]. However, the application of genomics-assisted breeding has received little attention in the development of saline-smart lentil cultivars. Dissanayake et al. (2021) identified candidate genes related to salinity tolerance, such as the high-affinity potassium transporters (HKT) and transcription factors on chromosomes 2 and 4, using the GWAS approach [50]. It was also observed that tolerant lentil accessions localize Na^+^ ions within the root tissues rather than transporting them throughout the plant under salt stress and maintain the same level of K^+^ ion concentration in the aerial part [50]. The results suggest that the Na^+^ absorbed by the plant was either actively transported back to the roots or retained in the root tissues to mitigate harmful effects on the shoots and leaves. Consequently, when considering potential candidate genes, HKT emerges as a key player that aligns with these findings. According to Roy et al. (2014), HKT is crucial for regulating Na^+^ ion transport [88]. In *Arabidopsis thaliana*, overexpressing HKT led to enhanced removal of Na^+^ ions from xylem tissues into specialized compartments within the roots, thereby preventing lethal effects on the plant [89]. The expression of HKT genes under salt stress has also been documented in various plant species, including rice and barley, where HKT was identified as the gene responsible for salt tolerance in both crops [90,91]. However, no SNP or allele sequences for the HKT gene have been generated for lentils under salt stress. Therefore, conducting allele resequencing of both tolerant and intolerant lentil accessions using an amplicon-based sequencing approach will be essential for analyzing the functional aspects of salt tolerance mechanisms in lentils. Singh et al. (2020) developed a mapping population by crossing salt-sensitive (L-4147 and L-4076) and salt-tolerant (PDL-1 and PSL-9) genotypes, which was used to screen genotypes against salt stress [47]. They tested the F_1_, F_2_, and F_3_ progenies of the parents for salinity tolerance in a 120 mM NaCl solution using seedling survival and the fluorescein diacetate (FDA) signal as criteria. It was discovered that the F_1_s were resistant to salt stress, demonstrating their superiority over the sensitive ones. The F_2_ segregation shows that a single dominant gene controls salinity stress tolerance. The F_3_ and backcross segregation data also supported these findings. The idea that the PDL-1 and PSL-9 tolerant genotypes were endowed with stress tolerance by the same gene was validated by the allelism test. This corresponded to the main QTL associated with seedling survival under salinity stress. The phenotypic variance of this trait was explained by the QTL located on linkage group 1 (LG_1) and mapped within a map distance of 133.02 cM in the F_2_ mapping population (L-4147 × PDL-1) [47]. Guan et al. (2014) and Lee et al. (2009) successfully mapped QTLs associated with seedling survival in soybeans, which were regulated by a single dominant gene [92,93]. However, other studies have reported conflicting findings, indicating that the mechanisms underlying salt tolerance are complex in nature [94,95]. Singh et al. (2021) identified 17,433 differentially expressed genes (DEGs) using a transcriptomics approach under salt stress [52]. DEGs were linked to cellular stress signaling, nitrogen metabolism, and phytohormone-mediated signaling. Singh et al. (2021) also identified hundreds of SNPs and 5643 SSRs that could be valuable for enhancing the lentil genetic linkage map under salt stress [52].

Various research studies in other legume crops have reported key genes and pathways associated with salt stress. A high degree of homology was observed between protein disulfide isomerases (PDIs) in *Cicer reticulatum*, *Lens culinaris*, *Phaseolus acutifolius*, *Pisum sativum*, *Oryza sativa*, and *Cicer arietinum*. Using existing in silico expression data for eight PDI genes, four of them—CPDI2, CaPDI6, CaPDI7, and CaPDI8—were found to be expressed in response to salt stress. It was observed that chickpea PDI genes are involved in salt stress tolerance, and the CaPDI genes may be further investigated for their potential to enhance salt tolerance [96]. Therefore, it is suggested that these identified genes and pathways in lentils be investigated, as they may be expressed under salt stress.

#### 2.1.3. Genome-Editing and Genetic Validation in Lentil

In crop genomics and breeding, identifying the genes encoding traits that are commercially important has proven to be a difficult undertaking. Recent developments in genetic analysis techniques and DNA sequencing technology have made it possible to identify several genes and hotspot genomic areas that influence desired features. Geneticists and plant breeders can improve crops, understand the molecular mechanisms of the genes behind desired qualities, and examine the genetics of complex mechanisms by identifying novel genomic areas or candidate genes.

GWAS and QTL mapping have dominated agricultural gene discovery research in recent times. To enhance understanding of crop functional genomics, investigations into the genetic basis of significant phenotypes are becoming routine. These studies yield information on underlying allelic variations, marker–trait associations, and the frequency of beneficial alleles in the target germplasm [97]. Nevertheless, additional verification is required for the identified loci prior to their potential application in breeding activities. The outcomes of many GWAS cases can be ambiguous due to challenges such as confounding population structures, low-frequency causal alleles that may lead to false-negative results, and unaccounted factors such as low-accuracy genotype calls at certain loci [98], as well as small population sizes [99,100]. Therefore, additional validation is essential, employing cross-population techniques where candidate loci are confirmed either in independent germplasm collections or in bi-parental populations [99].

The transcriptional profiling of genes like ADP-glucose pyrophosphorylase (AGPase) and pathogenesis-related protein 4 (PR4) in lentils under several physiological and biotic stress conditions has been performed using quantitative real-time PCR (qRT-PCR) [54,101]. AGPase, the primary regulatory and rate-limiting enzyme in starch biosynthesis, plays a crucial role in determining starch content, which in turn influences both yield and quality in plants [102,103,104]. Additionally, the expression of several AGPase genes in plants has been shown to be regulated by various stress conditions, including cold, salt, drought, and plant pathogens [104,105,106,107]. Therefore, it is recommended that AGPase genes in lentils be explored under salt stress conditions. While finding the most reliable reference gene is necessary for qRT-PCR studies, Saha and Vandemark (2013) have documented reference gene validation in lentils under cold stress and biotic stresses like *Sclerotinia sclerotiorum* and *Aphanomyces euteiches* infection [108]. The majority of earlier research on qRT-PCR analysis in lentils has either evaluated the stability of a single reference gene according to their experimental settings [101,109] or used a reference gene that has been verified in other species [54]. The expression stability of eight candidate genes: ribulose 1,5-bisphosphate carboxylase large subunit (Rbcl), ribosomal protein L2 (RPL2), 18S rRNA, tubulin (Tub), elongation factor 1α (EF1α), glyceraldehydes-3-phosphate dehydrogenase (GAPDH), heat shock protein (HSP), and Maturase (mat K)—was assessed in five lentil varieties at three distinct stages of leaf development and under abiotic stress conditions (salt, drought, cold, and heat) using qRT-PCR [55]. Singh et al. (2022) validated the identified ten genes under salt stress using qRT-PCR analysis (Proline dehydrogenase 2, MLP-like protein 423, Cyclin-dependent kinase G-2, Inactive poly [ADP-ribose] polymerase RCD1, Probable xyloglucan endotransglucosylase, Ferritin-2, Auxin transport protein BIG, ABC transporter G family member 25, Protein ZINC INDUCED FACILITATOR-LIKE 1, Probable serine/threonine-protein kinase SIS8) [110]. Despite the expression analysis, validation via advanced technologies such as KASP and CRISPR-Cas9 is negligible in lentil. Very few studies have been conducted on the development of KASP markers in lentil, and none have specifically addressed salt stress.

Breeding lentils can be enhanced through the use of genomic tools and technologies. There are ongoing gene transformation initiatives available for lentils. A notable example of such investigations is the transfer of the dehydration-responsive element binding gene (*DREB1A*), which is involved in plant responses to abiotic stresses, to lentils using Agrobacterium. This resulted in transgenic plants being resistant to salinity and drought [111]. Another crucial concern is in vitro regeneration following transformation. According to Sarker et al. (2003), a higher rate of shoot generation was observed when decapitated embryos were used compared to other tissues [56]. These endeavors offer valuable resources for future genome-editing investigations. A large number of potential genes linked to biotic and abiotic stress factors, as well as agronomic traits, have been identified in lentils [112]. Genome-editing technology may serve as a straightforward and cost-effective method to elucidate the function of candidate genes, facilitating the development of cultivars with desired characteristics such as stress resistance, including salinity [113]. However, to date, no genome-editing studies using CRISPR-Cas9, TALENs, or ZFNs have been conducted on potential genes in lentils.

The genomics investigations have greatly enhanced our understanding of salt stress adaptation by revealing key genes, QTLs, SNPs, and mechanisms associated with it. These results have defined new insights for precision lentil breeding strategies and provided information about the genetic basis of salt stress responses. Scientists can utilize this knowledge to develop saline-smart lentil varieties and ensure food security in the face of rapidly changing climatic conditions. In conclusion, research on the genetics of salt stress tolerance in lentil is still lacking, but recent results are illuminating key genes and mechanisms linked to salt stress. These results are critical to the development of resilient lentil cultivars that can withstand salt stress.

### 2.2. Lentil Transcriptomics

Transcriptomics essentially covers all techniques that analyze the transcriptional changes in plants. According to Afzal et al. (2020), transcriptomics successfully indicates the gene regulatory network and potential genes responsible for legume resistance to abiotic stress [114]. The utilization of serial analysis of gene expression (SAGE) and microarrays can provide comprehensive transcriptome information. The quantified gene expression assessment is performed using a relatively recently developed method called digital gene expression (DGE). Large-scale transcriptome data evaluation can be achieved at a low cost and high throughput using RNA-seq sequencing. Compared to microarray technology, this approach has a number of benefits, such as the ability to identify unique transcripts and the potential to generate probe sets without the need for genomic data [115].

Determining the genes linked to salt stress can help create elite genotypes and make it easier to manipulate traits and crops. Afzal et al. (2020) carried out whole transcriptome analyses of the salt-tolerant faba bean genotype (Hassawi-2) under varying salt stress concentrations [103]. The obtained DEGs encoded various regulatory and functional proteins, including kinases, plant hormones, transcriptional factors (TFs), basic helix-loop-helix (bHLH), Myeloblastosis (MYB), *WRKY*, HSPs, late embryogenesis abundant (LEA) proteins, dehydrin, antioxidant enzymes, and aquaporin proteins. Garg et al. (2016) conducted a whole transcriptome analysis on chickpea genotypes to explore the genetic underpinnings of drought and salinity stress response and adaptation [116]. Drought and salinity stresses had an impact on the major enzymes engaged in metabolic pathways, including protein modification, redox homeostasis, starch and sucrose metabolism, lipid metabolism, photosynthesis, carbohydrate metabolism, and the creation of precursor metabolites and energy. It is noteworthy that the members of the *AP2-EREBP* family demonstrated that certain transcript isoforms are expressed differently in different chickpea genotypes and/or developmental stages. According to Kavas et al. (2016), *BHLH* transcription factors are present in common beans that are salt-tolerant [117]. This finding could offer fundamental resources for studying the role of the bHLH protein in common beans and other species to improve agronomic and ecological production of economically significant bean crops. Hiz et al. (2014) conducted a comprehensive transcriptome study on root and leaf tissues of a tolerant common bean genotype cultivated in a hydroponic system under saline and control conditions, which aimed to discover DEGs and salt-associated GO terms and KEGG pathways [118]. Based on the GO and KEGG enrichment analyses of DEGs, it was demonstrated that root and leaf tissues control energy metabolism, transmembrane transport activity, and secondary metabolites in order to adapt to salinity. A total of 59 transcription factor families were used to classify 2678 potential common bean transcription factors, 441 of which were responsive to salt. Yu et al. (2016) used a transcriptome technique to examine the discovery of 188 soybean *WRKY* genes during salt stress [119]. The information presented here offers vital insights for additional functional research on the role of the *WRKY* gene in salt tolerance.

Advances in high-throughput NGS techniques have enhanced our understanding of salinity stress tolerance in various crops, such as wheat and tomato, by identifying salt-responsive genes through genomics and transcriptomics [120,121,122]. Transcriptomics can reveal the transcriptional structure of genes, their functional pathways, stress-associated critical transcripts, and other post-transcriptional alterations [123]. The molecular, morphological, physiological, and biochemical responses of salt-tolerant (PDL-1) and salt-sensitive (L-4076) lentil cultivars under control (0 mM NaCl) and salinity stress (120 mM NaCl) conditions at the seedling stage were analyzed through extensive transcriptome analysis. PDL-1 showed no signs of salt injury and had greater K^+^/Na^+^ ratios, relative water content (RWC), chlorophyll, glycine betaine, and soluble carbohydrates in leaves while having lower hydrogen peroxide (H_2_O_2_)-induced fluorescence signals in roots compared to L-4076 [52]. DEGs were found to be strongly associated with cellular redox homeostasis, secondary metabolism, nitrogen metabolism, phytohormone-mediated signal transduction, and cellular stress signaling. Singh et al. (2021) also identified 5643 SSRs and 176,433 SNPs that might be beneficial in the genetic mapping of traits under salt stress [52]. The identified salt stress-related pathways could be used for functional analysis in order to improve salt stress resistance in lentil [52]. The growth and yield-contributing characteristics of lentil significantly improved when supplemented with sodium nitroprusside (100 µM), particularly in plants cultivated under moderate salinity (50 mM NaCl). The administration of 100 µM sodium nitroprusside effectively protected lentil plants under moderate salinity by regulating plant development and metabolic pathways. Therefore, exogenous sodium nitroprusside application might be a helpful tactic for enhancing lentil plant performance in salinity-prone environments [124]. In conclusion, transcriptome analysis shows the dynamic variations in gene expression across many developmental stages and tissues. Thus, creating salt-tolerant lentil cultivars requires an understanding of these transcriptome dynamics.

### 2.3. Lentil miRNAomics

MicroRNAs, a unique family of short, non-coding regulatory RNAs, are primarily single-stranded and typically have a length of 20 to 24 nucleotides. They are essential for controlling the expression of certain genes in plants [125]. They cleave target transcripts or prevent translation in order to modify the expression of target genes [126,127]. These tiny regulators help plants resist abiotic stress by managing the regulatory network of multiple metabolic pathways in plants. They affect processes like protein refolding, the activation of antioxidant machinery, photosystem efficiency, and reproductive events [127,128,129]. While many plant species have thousands of miRNAs discovered, only a small number of studies have reported finding conserved stress-related miRNAs in lentils.

Six putative miRNAs identified as lcu-miR156, lcu-miR167, lcu-miR169, lcu-miR171, lcu-miR396, and lcu-miR390 belong to six conserved families and show differential expression levels in lentils under salt stress. The majority of biological activities, including differentiation, growth, signaling, and the response to biotic and abiotic stresses, depend on the target genes of the miRNAs that have been found. Additional reports indicate that certain transcription factors from the SBP, C2H2, GRAS, M-type_MADS, and MYB families are significantly influenced by the regulatory mechanisms of discovered miRNAs [130].

MiRNAs have the ability to control diseases, abiotic stress reactions, and plant growth and development. The hormone signaling crosstalk model of root growth and development in apple rootstock, *A. thaliana*, and Populus involves miR160, miR169, peu-miRn68, and 477b [131,132,133]. Cs-miR414 and cs-miR828 play a role in tea bud dormancy [134]. Pathogen stress regulation: miR397 negatively regulates apple resistance to the hepatitis B virus [135], miR396 modifies rice blast susceptibility [136], and miR528 boosts *Oryza sativa*’s capacity to defend against viruses [137]. MiR399 and miR827 play a significant role in the regulation of abiotic stress in terms of resistance to phosphorus deprivation [138,139]. According to Kawashima et al. (2009), the absence of sulfur causes the production of miR395 to regulate genes involved in the sulfur absorption pathway [140]. Yang et al. (2013) state that rice’s ability to withstand cold temperatures depends on the expression of miR319 [141]. MiR399 controls the timing of Arabidopsis flowering at various temperatures [142]. Recent research suggests that the miR169 family serves as a universal regulator of various abiotic stresses, as indicated by the comparative antagonistic expression profiles of miR169 [143]. Furthermore, findings suggest that miRNAs play a significant role in nutritional processes, as the overexpression of miR156 influences the expression levels of other miRNAs. This alteration leads to increased levels of anthocyanins, flavonoids, and flavonols while simultaneously decreasing the total lignin content [144].

These promising findings suggest the potential use of miRNAs to mitigate salt stress in lentils. Establishing a lentil miRNA network in response to salt stress will provide valuable insights into how to enhance salinity resistance through miRNA editing, ultimately aiding in the development of saline-tolerant cultivars.

### 2.4. Lentil Proteomics

Proteomics is an essential bridge between the transcriptome and metabolome, providing the protein composition of an organism at a particular point in time [145,146]. Proteomic techniques have made great progress in the last few decades as a potent next-generation research tool, particularly with the development of instruments with high mass accuracy and resolution [145,146,147]. These methods have been essential in elucidating the protein-level responses of plants to salt stress [147,148]. Proteomics contributes to the understanding of salt-stress tolerance through the identification of protein modulations and pathway changes [149]. Understanding the post-translational modifications of stress-induced proteins, which are essentially required by plants to adapt to different abiotic stresses, and the changes in proteins involved in vital biological pathways is made possible through proteomics [150,151]. Therefore, capturing the complete proteins produced in response to different abiotic stresses, such as salt stress, may be extremely important in advancing our understanding of the protein networks connected to salt-sensitive signaling pathways [152]. Proteomics holds significant value in understanding the plant’s diverse tactics to adjust to salt stress at the cellular, metabolic, and entire plant levels [149,153,154].

In response to salt stress, nucleoside diphosphate kinases (NDPKs) are activated, which reduces post-stress products and produces polyamines, antioxidant enzymes, and osmolytes to protect the plant from stress-related damage [155,156]. The abundance of *AtNDPK2*, which coordinates with MAPK-mediated H_2_O_2_ signaling, enhances resistance to oxidative stress and improves multiple stress tolerance by promoting the increased accumulation of various antioxidants through the induction of their respective genes [157]. NDPKs play a vital role in how plants respond to salt stress, operating through various mechanisms that include signaling pathways, regulation of gene expression, and metabolic adjustments. NDPK2 interacts with components of the (salt overly sensitive) (SOS) pathway, notably SOS2. This interaction is critical for adjusting plant responses to salt stress. Research indicates that double mutants lacking both NDPK2 and SOS2 show heightened sensitivity to salt, suggesting that NDPK2 is integral to the signaling cascade and is responsible for mediating salt tolerance [158]. NDPK2 also plays a significant role in regulating H_2_O_2_ levels, which are essential for signaling during stress responses. The double mutant exhibited altered patterns of H_2_O_2_ accumulation, indicating that NDPK2 influences not only ion transport but also the overall oxidative stress response [158]. Furthermore, nucleoside diphosphate kinase B (NDPKB) is an enzyme that converts GTP to ATP and participates in the H_2_O_2_-mediated mitogen-activated protein kinase signaling pathway. Research has shown that NDPK enhances tolerance to NaCl in various plant species, including Arabidopsis, creeping bentgrass, and rice [159,160]. Superoxide dismutase (SOD) increases the plant’s resilience by eliminating ROS and lessening the oxidative damage caused by salt stress to the cells [161]. Proteomics under salt stress reported key genes and pathways in other legume crops that help us to deeply understand salt stress mechanisms and networks. Such discoveries might be beneficial for developing saline-smart lentil cultivars. Comprehensive proteome investigations are primarily available for two important legume crops, namely *M. truncatula* and soybean [162]. Salinity stress induces changes in key proteins, including chaperonin protein, fructose-bisphosphate aldolase, and HSPs. These proteins are linked to various processes, such as energy and carbohydrate metabolism, signal transduction pathways, antioxidant/ROS scavenging, and photosynthesis [163]. It was shown that the response of soybean leaves, hypocotyls, and roots to salt stress resulted in an abundance of proteins involved in photosynthesis. According to Sobhanian et al. (2010), proteins such as “glyceraldehyde-3-phosphate dehydrogenase (GAPDH)” and “fructokinase (FRK)” are abundant in leaves or hypocotyls, and hypocotyls or roots, respectively. Both proteins are involved in photosynthesis and other metabolic processes under salinity stress [164]. Recent studies have indicated that, beyond its role in glycolysis, plant GAPDH also has non-glycolytic functions related to abiotic stress. For instance, overexpressing *OsGAPC3* in rice led to enhanced salt stress tolerance in transgenic lines, as *OsGAPC3* can mitigate salt toxicity by regulating H_2_O_2_ levels. Additionally, in aspen, expression of the GAPDH gene was found to be upregulated in response to drought stress [165]. In wheat, the expression of *TaGAPDH12* was significantly increased in shoots following exposure to four types of abiotic stresses: cold, heat, salt, and drought [166]. In Arabidopsis, *GAPC1* and *GAPC2* engage with plasma membrane-bound phospholipase D (PLD) to regulate the plant’s response to abscisic acid (ABA) and water scarcity [167]. In soybean, transgenic plants overexpressing *GmGAPDH14* demonstrated improved growth compared to the control after being subjected to salt stress [168]. The overexpression of *PsGAPDH* has been shown to enhance salt tolerance in potato [169]. In Arabidopsis, the overexpression of the *TaGApC* gene from the Chinese spring variety of wheat resulted in enhanced drought tolerance by reducing levels of ROS [170]. In summary, members of the GAPDH gene family have been shown to be associated with drought, heat, cold, and salt stress, with their expression levels varying across different tissues and developmental stages [171]. While GAPDHs have been characterized and studied in numerous plant species, the characterization and the role of GAPDH in regulating the molecular mechanisms of salt stress in lentils is still unknown. Fructose must initially be phosphorylated to fructose-6-phosphate by FRKs or hexokinases (HXKs) before it can proceed with further metabolism [172]. The affinity of FRKs for fructose is significantly higher than that of HXKs, indicating that fructose in plants is likely primarily phosphorylated by FRKs [173,174]. Plant FRKs are part of the pfkB family of carbohydrate kinases, characterized by a di-gly (GG) motif at the N-terminal region and a GAGD motif at the C-terminal region [175]. Different plant species possess between two to eight FRK genes. FRK genes have been cloned from a variety of species, including tomato, rice, mandarin orange, Arabidopsis thaliana, maize, and aspen [175,176,177,178,179]. Twenty-two differentially regulated cowpea proteins under salt stress were isolated from two contrasting genotypes [56]. Crucial proteins involved in energy metabolism and photosynthesis, such as rubisco activase, ribulose-5-phosphate kinase (Ru5PK), and oxygen-evolving enhancer (OEE) protein 2, were abundant in genotypes resistant to salt stress. In contrast, these crucial processes adversely affect salt-sensitive cultivars, restricting their growth [56]. Using MALDI-TOF/TOF mass spectrometry, 43 responsive proteins were identified that are related to changes in cellular metabolism in salt-stressed soybeans. Under salt stress, a total of 43 proteins were observed, of which six were found to be unique [180]. A proteomic investigation conducted at different time intervals revealed 2692 phosphoproteins and 5509 phosphorylation sites in contrasting soybean cultivars under salt stress, providing significant insights into the role of phosphoproteins in salt tolerance [181]. Plants have developed a range of adaptive mechanisms to detect and respond to salinity signals, which involve multiple phosphorylation cascades. These include the SOS pathway, the phosphatidic acid (PA)-mediated activation of calcium-dependent protein kinases (CDPK), and the ABA-regulated activation of mitogen-activated protein kinase (MAPK) cascades [182,183,184,185]. The phosphorylation of specific signaling components is known to occur at critical time points after plants are exposed to salt stress, and these modifications coordinate particular metabolic processes, cell wall porosity, and lateral root initiation, aiding plants in their adaptation to salt stress [183,184,186]. The SOS pathway is essential for interpreting calcium signals triggered by salt stress. It includes proteins like SOS3, which detects calcium levels, and SOS2, which is activated via phosphorylation. This pathway plays a key role in maintaining ionic balance by regulating the Na^+^/H^+^ antiporter SOS1, which is vital for the exclusion of sodium ions [187]. The transcription factor INDETERMINATE DOMAIN 4 (IDD4) undergoes phosphorylation by MPK6 in response to salt stress, which enhances its capability to regulate genes involved in stress responses. Mutations in IDD4 that disrupt phosphorylation can result in increased salt tolerance, highlighting its function as a central regulator of transcriptional responses to salinity [188]. Furthermore, it was shown that the phosphorylation of certain TFs, such as *MYB*/*MYB* TF-like proteins, controls the salt-responsive gene or genes that are involved in chalcone metabolism (chalcone synthase). The iTRAQ (Isobaric Tags for Relative and Absolute Quantitation) method contrasts sharply with older techniques such as two-dimensional gel electrophoresis (2-DE), which typically analyzes one sample at a time. This results in higher throughput and minimized variability in results. iTRAQ enables accurate quantification of protein abundance, making it especially advantageous for studying differential protein expression under salt stress. Compared to traditional methods, iTRAQ offers enhanced sensitivity and broader proteome coverage. Additionally, this technique produces extensive datasets that encompass not only protein identification but also quantitative data across various conditions [189,190]. Köse (2012) screened salt-resistant and salt-sensitive lentil cultivars at the protein level using SDS-PAGE [58]. They noted that stress treatment reduced the strength of various proteins within each cultivar. The SDS-PAGE did not provide sufficient information related to the proteins under salt stress. It was suggested that two-dimensional electrophoresis, iTRAQ test, and LC-MS/MS analysis should be used for screening lentil germplasm at the protein level rather than SDS-PAGE. The iTRAQ test was used to identify 278 and 440 salt-responsive proteins from the roots and leaves of soybeans under salt stress [152]. All of the identified proteins fall into one of thirteen categories, which include signaling, stress and defense, glucose metabolism, and membrane transport to content courtesy of cell division. Moreover, Aghaei et al. (2009) found that the hypocotyls and root of soybean under salt stress showed lectin proteins, protease inhibitor proteins, late embryogenesis-abundant (LEA) protein, beta-conglycinin, elicitor peptide-3 precursor, and basic/helix-loop-helix protein [191]. Ngara and Ndimba (2014) also enhanced our understanding of the underlying candidate genes, intricate regulatory systems, and key signaling actors involved in salinity tolerance and response in legume crops [192]. Seedlings of two distinct chickpea genotypes, Flip 97-43c (salt tolerant) and Flip 97-196c (salt sensitive), showed 364 reproducible spots after ten days of salt treatments. The identified proteins were found to be associated with photosynthesis (39%; viz. chlorophyll a-b binding protein, oxygen-evolving enhancer protein, ATP synthase, RuBisCO subunits, carbonic anhydrase, and fructose-bisphosphate aldolase), stress responsiveness (21%; viz. HSP, chaperonin, LEA-2, and APX, as well as protein synthesis and degradation (14%; viz. zinc metalloprotease FTSH 2 and elongation factor Tu) [193]. The expression of LEA genes is controlled by various cis-acting elements that respond to environmental stresses. One notable promoter region is the abscisic acid-responsive element (ABRE), which plays a crucial role in boosting the transcription of LEA genes during salt stress [194]. Furthermore, microRNAs have been shown to influence the expression of LEA genes, thereby connecting genetic regulation to stress response mechanisms [194]. Recent research suggests that post-translational modifications (PTMs), particularly phosphorylation, play a significant role in the functionality of LEA proteins under salt stress. These modifications can affect protein stability and activity, thereby enhancing their protective functions during exposure to salinity [195,196]. Such research work is highly suggested to be conducted in lentil in order to undermine the key genes, mechanisms, and pathways involved in salt stress tolerance.

At an EC of 3 dS/m (~30 mM NaCl), lentils have a yield loss of approximately 90%, making them salt-sensitive legumes compared to other crops [63]. Lentil seeds exhibited a progressive reduction in germination when transferred to water after 16, 40, and 64 h of exposure to 0.33 M NaCl, following an initial 8 h of water imbibition. Two-dimensional electrophoresis separation and computer image analysis of protein spots both demonstrated that these salt-related changes were accompanied by changes in the protein patterns of the embryo axis. Proteins have been tentatively identified by comparing the protein maps of germinating Arabidopsis seeds with those of lentil embryos imbibed under non-lethal salt stress [57]. They characterized proteins associated with the later stages of germination, including LEA proteins, HSP, elongation factors, enzymes involved in oxidative stress, and α- and β-tubulin. This diverse proteomic response to salt stress may be attributed to the accumulation of mRNAs and related proteins, as observed in the roots of salt-tolerant crops. The detection of HSPs in the 2–3 mm embryo axes of germinating seeds pre-treated with NaCl suggests that the accumulation of HSPs may play a role in helping plants cope with hostile environmental conditions, potentially serving as biochemical markers for germination [57]. Similar findings have also been reported in other legume crops, such as pea and soybean, where LEA proteins, elongation factors, HSPs, and oxidative stress-related enzymes were identified under salt stress. Under salt stress, the expression of HSP genes is markedly increased. Research has shown that proteins such as Hsp17.8, Hsp26.3, Hsp70, and Hsp101 are produced at elevated levels in plants exposed to NaCl stress, underscoring their role in stress response mechanisms [197]. The transcriptional activation of these genes is primarily regulated by heat shock factors (HSFs), which bind to specific promoter regions of HSP genes in response to stress signals [198]. HSPs primarily serve as molecular chaperones, preventing protein aggregation and misfolding that can arise under salt-stress conditions. HSPs serve a protective function by acting as scavengers of ROS, thereby reducing oxidative damage. Additionally, they collaborate with antioxidant enzymes to bolster the plant’s capacity to withstand oxidative stress caused by high salinity [195]. HSPs do not function independently; they engage with other stress-related proteins and signaling pathways. For instance, they are involved in signaling mechanisms mediated by phytohormones like ABA, which is crucial for plant responses to abiotic stresses, including salinity [199,200]. The interaction between α- and β-tubulin is primarily stabilized by electrostatic attractive forces, which are essential for the formation of heterodimers. These interactions are stronger than those observed in α/α or β/β tubulin complexes, suggesting that salt may impact these dynamics by modifying ionic conditions that influence electrostatic interactions [200,201]. Each tubulin monomer contains nucleotide binding sites; GTP is bound to α-tubulin, while β-tubulin can bind either GTP or GDP. The hydrolysis of GTP to GDP during microtubule assembly is crucial for dynamic instability, a process that can be affected by high salt conditions [201,202]. The presence of GTP stabilizes the microtubule structure, whereas its absence results in depolymerization. Overall, proteomics has offered valuable, data-driven insights into the genetic basis of salt tolerance by identifying the molecular drivers and pathways associated with these traits. In particular, several protein families—including NDPK, GAPDH, FRK, HSP, and LEA—have been extensively studied by various sources and are recommended as focal points for future lentil proteomic research. This focus will ultimately contribute to the development of lentil cultivars that exhibit greater resistance to salt stress.

### 2.5. Lentil Metabolomics

Metabolomics has emerged as an effective omic approach in the study of plant stress, providing new insights into the complex signaling and plant adaptation mechanisms to deal with a variety of environmental stresses [203]. Plant physiological and biochemical responses to abiotic and biotic stresses can be better understood using metabolomic techniques that can produce a detailed profile of the metabolites found in plant tissues [204]. The knowledge gathered from metabolomic research holds great promise for improving crop yields since it can direct the development of more resilient and stress-tolerant genotypes through genetic engineering as well as targeted breeding [205]. Metabolomics can aid in selecting superior genotypes and optimizing cultivation techniques to enhance plant performance in harsh climatic conditions by identifying metabolic signatures associated with desired traits [206]. Salinity stress is also linked to changes in biochemical activities such as proline, soluble sugar, soluble protein, chlorophyll content, and the activities of CAT, proline dehydrogenase (PDH), and ascorbateoxidase (AO) [207].

Metabolomic investigations have demonstrated that lentils undergo significant metabolic modifications in response to salinity stress. The leaves and roots of lentil genotypes show a decrease in organic acids. In contrast, the roots show an enhancement of important metabolites such as L-asparagine, D-trehalose, allantoin, and urea, while the leaves accumulate sugars and polyols. Furthermore, it has been observed that the leaves accumulate toxic Cl ions, suggesting that this could be a protective mechanism against salt stress, with the particles being segregated within the vacuoles. The investigation also proposes a scenario in which legumes regulate a metabolic pathway involving purine catabolism to obtain restricted amounts of carbon and nitrogen under salt stress [208]. Several important metabolic pathways and putative biomarkers associated with lentil salinity tolerance have been identified through metabolomic research.

For the first time, Gaafar and Seyam (2018) provide insights into the ascorbate-glutathione cycle’s role in lentil salt tolerance, offering valuable information for breeding saline-tolerant cultivars [58]. Lentil genotype Giza 9 (salt-tolerant) was observed to enhance its germination, water content, and proline under salt stress, which was associated with an active ascorbate-glutathione cycle. Giza 9 roots exhibited an elevated expression of antioxidant defense genes [GR, APX, SOD, aldehyde dehydrogenase (ADH), and CYS3), whereas Giza 4 showed higher expression of only the CAT gene. Higher levels of GR and APX activity in Giza 9 roots provided additional evidence. Studies on metabolites have shown that the genetics of lentil genotypes also play an important role in their response to salt stress. Muscolo et al. (2015) investigated the responses of four different lentil genotypes (Eston, Pantelleria, Ustica, and Castelluccio di Norcia) to salt and drought stress during germination and early growth stages [59]. It turned out that genotype Pantelleria showed more resistance to salinity stress than genotype Eston. Notably, all cultivars showed a reduction in the amounts of tricarboxylic acid cycle intermediates under salinity stress. By comparing the sensitive variety (Eston) to the salt-tolerant (Pantelleria), the investigation showed a greater decrease in threonic acid levels. Furthermore, alanine and homoserine were proposed as putative metabolic markers for salt stress [59]. Al-Quraan and Al-Omari (2017) reported that the GABA (γ-aminobutyric acid) shunt is a key signaling and metabolic pathway that allows the adaptation of lentil seedlings to salt, osmotic, and oxidative stresses using two lentil cultivars (Jordan 1 and Jordan 2) [60].

Previously, many researchers used iso-osmotic solutions containing various solutes (NaCl and KCl salts, as well as PEG) to apply distinct stresses and monitor the unique responses of lentil seedlings. The plants’ different characteristics, such as height, chlorophyll content, proline levels, H_2_O_2_, reduced ascorbate (AsA), total GSH, and ion uptake during seedling development, were significantly altered by these applied solutions. Additionally, the solutions affected the activities of several antioxidant enzymes, including GR, APX, DHAR, CAT, but not MDHAR [61]. Insights gained from these metabolomic studies can aid in developing lentil cultivars capable of withstanding salinity and enhancing productivity in saline environments.

Metabolomics, a novel paradigm, has emerged as a promising method to identify signature metabolites as indicators linked to significant features [209]. Metabolite levels are regarded as quantitative phenotypic traits in QTL mapping and GWAS due to their quantitative nature. Although metabolite-based QTL mapping (mQTL) and GWAS (mGWAS) are commonly used for genomic mapping of metabolites, no study on lentils under salt stress has been conducted yet [210]. Thus, extensive research on metabolomics-assisted breeding is necessary to effectively screen and select breeding material for increased salt stress tolerance and enhanced yield in lentils.

### 2.6. Lentil Phenomics

In the post-genomic era, phenomics, the study of plant phenotype, has gained significance for understanding how plants respond to various environmental stress conditions, including salt stress [211]. Several methods were commonly employed to investigate plant phenomes, including integrated imaging, visible light, infrared, thermal-based, fluorescence, and spectroscopy [211]. Several investigations have successfully employed phenomics to study how plants react to salt stress.

High-throughput phenotyping platforms, such as The Plant Accelerator in Adelaide, Australia, and the International Plant Phenotyping Network (http://www.plant-phenotyping.org/) (accessed on 26 August 2024), had recently been established to improve the precision and throughput of trait phenotyping, including response to salinity. Next-generation phenotyping techniques offer several advantages over traditional methods, including spectral imaging of complex traits, automated data collection, non-destructive and non-invasive measurement of phenotypes, and the ability to generate precise records such as “ionic responses” under salinity stress [19,212,213,214]. Image-based phenotyping is regarded as one of the high-throughput phenotyping platforms that observe small variations in morphological and physiological traits at high time and spatial resolution [19]. Other responses include chlorophyll fluorescence, leaf water content, relative growth rate, and leaf senescence under salinity stress observed across a vast collection of germplasm or mapping populations simultaneously [212,213,215,216].

In this context, it is noteworthy that the dynamics of relative growth rate under salt stress in chickpea were evaluated at various time scales using a high-resolution imaging system equipped with a fixed 5-megapixel visible/RGB camera [217]. High throughput phenotyping in plants allows for precise evaluation of the effects of salt stress at different physiological levels, such as photosynthesis, transpiration, ionic relations, plant senescence, as well as on yield and other traits related to salinity tolerance [19]. Technological developments in plant phenomics, such as automated and digital imaging, could expand our knowledge of the many temporal responses of genotypes under saline stress. To improve the accuracy of recognizing, quantifying, and predicting plant salinity responses, deep learning [211], active vision cell (AVC) image-acquisition [218], and other contemporary phenotyping technologies could be utilized.

Although implementing phenomics technologies in lentils has proven difficult, phenomics offers an interesting way to explore plant behavior in relation to environmental interactions based on genetic backgrounds. This is mainly because lentils show an entirely different phenomenon in controlled stress conditions compared to their behavior in the field [87]. Therefore, it is suggested that phenomics be utilized in field-based lentil research. Traditional phenotyping methods have undergone a number of changes in order to capture the physiological responses in lentils.

The current approach to understanding the phenotyping of different traits in lentils is labor-intensive due to a lack of trained staff. Other major drawbacks of this method include low efficiency and the need to develop a reliable scoring system. High-throughput phenotyping strategies, on the other hand, may be able to solve the issues with traditional phenotyping techniques by facilitating the quick measurement of complex traits like plant growth, yield in both controlled and field conditions, and resistance to biotic and abiotic stresses [46]. High-throughput phenotyping has become more popular in a number of crop plants, such as pulses and cereals [21], while it is still rarely used in lentils [215,217,219,220,221]. Dissanayake et al. (2020) created a phenotyping technique based on RGB imaging to screen lentils for salt toxicity [46]. High-throughput phenotyping techniques exhibit more precision and accuracy than conventional methods. As a result, they may be utilized in lentil molecular breeding projects to effectively identify the locus(s) influencing desired traits. Dissanayake et al. (2020) conducted the only study that has created a high-throughput phenotyping method for screening lentils under salinity [46]. A moderate association (r = 0.55; *p* < 0.0001) was found between the salt tolerance scores derived from conventional screening and image-based assessment. This association was further validated through Spearman rank correlation analysis (r = 0.68; *p* < 0.0001). It is interesting to note that the lentil line CIPAL1522, according to traditional phenotyping, was rated as tolerant, but the high-throughput phenotyping approach found it to be moderately tolerant. Detailed phenotypic trait assessments were used to compare the high-throughput phenotyping and conventional screens. The results showed that high-throughput phenotyping provided better precision and consistency than conventional phenotyping. Phenomics has the potential to expedite efforts to improve salt tolerance in lentil breeding, even though it is still in its infancy in understanding the effects of salt stress on lentils [222]. The development of saline-smart lentil genotypes can be enhanced, and valuable insights can be obtained by integrating omics technologies with phenomics [223].

### 2.7. Lentil Epigenomics

Over the years, the genetic diversity present in crop plants has been primarily utilized to improve several agronomic features, including salt tolerance in crop plants. This has frequently involved using genetic variation concealed across wild varieties/species of crop plants. However, the sources of genetic variation have steadily decreased due to the loss of numerous crop plant natural habitats and the constant use of natural genetic resources [224,225]. One of the main obstacles to plant breeding nowadays is the lack of genetic diversity for many crop species, including lentils.

A variety of chemical mutagens and radiation treatments have been employed to induce genetic diversity in order to solve this issue. Despite efforts to increase genetic variation, it has not been possible to produce desirable alleles for a number of traits at the necessary rate. Accelerated creation of genetic diversity seems to be a major need in order to address rapidly changing soil and climatic circumstances. In addition to taking large amounts of time, traditional crop enhancement techniques also have other obstacles. First of all, because undesirable and desirable alleles are closely related genetically, it is quite difficult to distinguish between them. Secondly, certain chromosomal regions function as barriers to cross-over, reinforcing the connection between numerous loci/alleles. Existing approaches include screening a large number of individuals to find appropriate segregants, but because this method is labor- and time-intensive, it is not always feasible. In order to induce heritable epigenetic variation in lentils for the improvement of traits of significance, such as salt tolerance, we propose and describe novel ways based on recent advances in the field of epigenetics/epigenomics [226]. The described epigenetic variants are anticipated to be useful tools for enhancing lentil salt tolerance, as numerous studies have reported the involvement of modulated DNA methylation and histone modifications in salt stress tolerance [227].

The various aspects of epigenetics in legumes have been the subject of increased research over the last few years [228,229]. Although the majority of previous research only provided data on specific loci, developments in high-throughput sequencing methods have made it possible to profile epigenetic modifications across the entire genome. The role of histone modifications such as H3K4me3, H3K9ac, and H3K27me3 in nodulation was examined using chromatin immunoprecipitation (ChIP) sequencing (ChIP-seq) or ChIP-quantitative polymerase chain reaction (ChIP-qPCR) [230]. Whole-genome bisulfite sequencing was also used to examine the methylation pattern in soybean cotyledon during seed maturity. An et al. (2017) found a correlation between the differential expression of 77 seed-specific genes and differential CHH methylation [231]. Epigenetic control is also involved in legume responses to abiotic stress. Stress has been shown to globally alter epigenetic characteristics. For instance, chickpea exhibited an overall increase in histone 3 lysine 9 (H3K9) acetylation under salt and drought stress [232]. In the meantime, common beans under cold stress also showed increased expression of histone deacetylase genes [233]. Drought-treated faba beans and barrel clover exhibited a genome-wide decrease in DNA methylation [234,235]. Table 2 presents a list of the epigenetics research work on other legume crops. These findings suggest that significant alterations in epigenetic characteristics play a key role in legume stress responses. No study focusing on epigenetics has been conducted on lentils so far. Therefore, comprehensive research is needed to address the roles of epigenetic components, including DNA methylation, histone modifications, and noncoding RNAs in lentil under salt stress.

### 2.8. Lentil Ionomics

The study of ionomics is crucial for comprehending how plants take up, transport, and retain nutrients from the soil—particularly when they are subjected to environmental stresses like salinity. The pool of necessary inorganic nutrients that plants need in very modest amounts for metabolism and stress adaptation is known as the ionome [244,245]. High-throughput elemental composition profiling and variations in response to different stimuli and/or stressful conditions are examples of ionomics techniques [244,245].

Salinity has an impact on plants’ competitive nutrient intake, accumulation, and transport. In the presence of salinity, plants experience nutritional imbalance caused by high concentrations of Na^+^ and Cl^–^ ions in the rhizosphere, which can disrupt the absorption of other nutrients such as nitrogen (N), phosphorus (P), potassium (K), boron (B), calcium (Ca), zinc (Zn), copper (Cu), magnesium (Mg), and iron (Fe) [246,247]. Previous studies have verified that an ionic imbalance, specifically involving K^+^ and Ca^2+^, may be the cause of the detrimental effects of salt stress on the plant. If plants are to thrive in a salt-stress environment, they need to retain relatively greater amounts of K^+^ and Ca^2+^ [248]. Elevated salt concentrations decrease the levels of Ca^2+^, K^+^, Mg^2+^, and other cations, all of which are essential to plant photosynthetic activity [249]. For example, a notable decrease in the Na^+^/K^+^ ratio was noted, which was caused by the competing absorption of the Na^+^ and K^+^ ion flows. This led to a K^+^ deficit and notable yield losses [250].

Yerli Kırmızı and Fırat87 demonstrated moderate salt stress tolerance, whereas Altın Toprak and Çağıl were recognized as lentil genotypes with high salt tolerance [52]. Earlier research [59,63,251] found three cultivars that could withstand varying salinity concentrations in southern and central Italy: two cultivars called Ustica (UST) and Pantelleria (PANT) in the homonymous islands and Castelluccio di Norcia (CAST) in central Italy. Singh et al. (2017) tested 162 lentil genotypes and found that in comparison to the sensitive cultivar (L-4076), the root and shoot anatomy of the tolerant line (PDL-1) and wild accession (ILWL-137) displayed restricted uptake of Na^+^ and Cl^–^ due to thick layers of their epidermis and endodermis [48]. According to Singh et al. (2020), the salt-tolerant lentil cultivar PDL-1 was able to withstand the adverse effects of salt stress by maintaining a higher K^+^/Na^+^ ratio in the cytosol [47]. Additionally, salt-tolerant cultivars exhibited higher RWC, chlorophyll, glycine betaine, and soluble sugar levels and lower fluorescence signals (indicating reduced H_2_O_2_ generation), which suggested less Na^+^ accumulation in leaf tissues [47]. According to Panuccio et al. (2022), resistant lentil genotypes employ strategic mechanisms, including increased proline accumulation, reduced sodium uptake, and modified photosynthetic characteristics to mitigate salt stress [64]. The findings of Singh et al. (2017), Singh et al. (2020), and Panuccio et al. (2022) reported overlapping results of restricted uptake of Na^+^ and Cl^-^ due to thick layers of epidermis and endodermis and maintaining a higher K^+^/Na^+^ ratio in the cytosol of tolerant genotypes under salt stress [47,48,64]. Exogenous sodium nitroprusside has been shown to alleviate salt stress in lentils [113]. Growth traits were significantly suppressed by salinity stress; however, MDA, H_2_O_2_, SOD, CAT, and peroxidase contents were elevated. The characteristics that affect growth and yield were significantly improved in plants provided with sodium nitroprusside. Sodium nitroprusside was nevertheless a useful treatment for lentil plants in moderate salinity because it controlled plant development and metabolic processes. Therefore, it may be possible to improve the performance of lentil plants in salinity-prone environments by applying sodium nitroprusside exogenously. Hossain et al. (2017) highlighted the significance of utilizing iso-osmotic solutions to enhance our comprehension of salt stress in lentils [61]. They used NaCl, KCl, and PEG as sources of salt stress. The results indicated that NaCl induced greater susceptibility to salt stress compared to iso-osmotic solutions of KCl and PEG. This was evidenced by leaf chlorosis, decreased K^+^ levels, disrupted ion homeostasis, and increased levels of MDA, H_2_O_2_, and proline. When Ca and NaCl were added, there was no chlorosis, and the K^+^ content was higher. Additionally, it was found that lentils are highly vulnerable to K^+^ leakage when exposed to NaCl stress. Also, achieving salt tolerance in lentils necessitates a reduction in K^+^ leakage. Trehalose-functionalized silica nanoparticles (TSiNPs) can improve lentil response to salt stress [252]. The growth of seedlings (shoot and root length), ionic balance (K^+^/Na^+^ ratio), and osmolyte status (sugars, proline, glycine betaine, trehalose) were all improved by SiNPs (silica nanoparticles) and TSiNPs. Furthermore, increased antioxidant enzyme activities helped in scavenging ROS produced in seedlings stressed by NaCl, thereby enhancing membrane integrity (by reducing MDA and EL). TSiNPs showed significantly greater efficacy in mitigating salt stress compared to SiNPs.

The majority of research on salinity stress in lentils has been conducted under controlled conditions at the seedling stage and in vitro during germination [48,253,254]. Few studies have examined the development and performance of lentils in open fields when the salinity is greater than 5 dS/m. Additionally, there have not been many studies performed on lentil ionomics, and more research is needed to fully utilize the genes and pathways involved in salt stress. Even though ionomics is more frequently linked to researching salinity responses, it is advised to carry out more research on lentils under salt stress in order to create saline-smart cultivars.

### 2.9. Enhancing Lentil Breeding Through Single-Cell Omics

Advances in technology have enabled scientists to examine thousands of cells and multiple molecular dimensions—such as the genome, transcriptome, proteome, ionome, metabolome, and epigenetic modifications—at high resolution using single-cell (sc) methods [255,256,257]. Additionally, several databases such as PlantscRNAdb [258], Single-Cell Portal [259], PsctH [260], PCMDB [261], IonFlow [262], scDEC [263], RA3 [264], epiAnno [265], and PlantCADB [266] have been developed to facilitate the application of single-cell techniques in plant biology. These databases, combined with computational biology, have opened up possibilities for single-cell genome-wide approaches to assess various molecules—including chromatin, DNA, RNA, and protein—in many organisms, including plants, at the highest resolution [255,256,257]. For example, Argelaguet et al. (2021) mentioned analyzing the sc-genome, methylome, or chromatin accessibility using genomic DNA [261]. It is possible to profile the transcriptome (RNA) and proteome (proteins) of a single cell.

Key genes and pathways in various plant species have been reported by prior research using sc, utilizing the omics tools of genomics and metabolomics [267,268,269,270,271]. Therefore, it is possible to detect genes, metabolites, proteins, ions, and epigenetic elements in lentils under salt stress at the sc-level by using sc-omics analysis. Furthermore, sc-omics has enormous potential for breeding, genetic research, and pathway engineering for salt stress, as no sc-omics study has been reported in lentil to date.

### 2.10. Accelerating Lentil Breeding Through Machine Learning

Phenotypic data collection is a crucial initial step to classify crops as either sensitive or tolerant to specific stresses [9,272,273]. The labor-intensive, time-consuming, potentially destructive, subjective, expensive, inefficient, and lack of inter- or intra-rate repeatability associated with manual phenotyping has resulted in a demand for effective, automated, and precise technologies capable of gathering phenotypic data across all growth stages and correlating it with genomic information [274]. The primary challenge hindering crop breeding efforts has been effectively addressed by the development of high-throughput phenotyping combined with artificial intelligence and machine learning [8,9,10,275].

The integration of machine learning with sc-RNA-seq has been growing [267]. This is especially true when analyzing unsupervised sc-transcriptomes to reconstruct sc-developmental trajectories over pseudotime, which has led to the identification of hundreds of genes that express differently in different types of cells [267]. Most sc-RNA-seq and metabolomics research has documented the grouping of hundreds to millions of sc-genes, primarily by the application of unsupervised machine-learning techniques [267,276]. Pan-genome construction [277,278,279] is facilitated by machine learning, enabling the identification of core, dispensable, and specific genes that accelerate functional validation and uncover regulatory roles in genomics [235,280]. Machine learning facilitates pan-genome building [277,278,279], allowing for the discovery of core, dispensable, and particular genes that speed functional validation and reveal regulatory functions in genomics [280,281].

Moreover, machine learning algorithms have been applied to precision agriculture, automated irrigation, as well as crop yield, and complex trait prediction [282,283,284,285]. Additionally, genomic selection and the identification of genomic regions associated with specific traits are made feasible by machine learning [286,287,288]. Furthermore, it can translate biological knowledge and data into precision-designed plant breeding by advancing omics sciences in plant biology and accelerating the discovery of agronomically valuable genes, mutations, and metabolites for knowledge-driven molecular breeding [10,289]. Machine learning approaches have been utilized in commercial breeding operations to create predictive models for data-driven genomic design breeding [10]. Its capacity to recognize predictive patterns enhances conventional comparative genomics methodologies in plant science [289]. Machine learning has been applied to comprehend and predict gene expression in plants under biotic and abiotic stresses, as well as regulatory architecture [290,291]. It is anticipated that machine learning will continue to play a crucial role in leveraging the rapidly accumulating multi-omics data in plant biology for the development of stress- and salt-tolerant lentil cultivars. Current challenges in plant stress research, particularly in salt-related phenotyping, can be addressed by automated phenotyping systems that integrate machine learning and artificial intelligence [292].

There is a scarcity of information on the implementation of machine learning and artificial intelligence in lentils. Naik et al. (2024) documented EfficientNetB0 as the best model among the 18 CNN models utilized in lentils [293]. Butuner et al. (2023) use machine learning techniques to categorize lentil images [294]. The ANN method produced a maximum classification success rate of 99.80% using the deep features that were derived from the SqueezeNet model. The findings also demonstrated that the use of grain size and shape parameters in image classification can produce more accurate and detailed results than is actually possible when using manual quality assessment. Aasim et al. (2023) employed a multilayer perceptron (MLP) model to assess and predict the salt tolerance index (STI) values and in vitro growth parameters in chickpea [295]. The results showed that chickpea under salt stress had relatively high R^2^ values. While machine learning and AI methods have been applied to other salt-stressed crops, no study on lentils has been published yet. Consequently, cutting-edge machine learning and artificial intelligence techniques are strongly advised to be applied in order to create saline-smart lentil cultivars.

### 2.11. Advancing Lentil Cultivars Development Through Speed Breeding

The idea of speed breeding has proven to be a useful tool for accelerating the creation of new cultivars for a wide range of applications [11,12,13]. Speed breeding approaches include manipulating the environmental conditions in which they are grown in order to promote crop genotypes to the next breeding generation as soon as possible [296,297]. This causes faster flowering and an earlier seed set. It normally takes 8 to 12 years in conventional breeding for annual crops like maize, barley, wheat, and soybeans, from the crossing of parental lines to the commercial release of new cultivars [298]. The time required for breeding can be reduced by half or more through speed breeding [296].

Accelerated breeding techniques such as genome editing, genomic prediction, and omics-assisted selection can be combined with or used alongside speed breeding to develop improved cultivars [298]. For example, Rana et al. (2019) employed a biotron-based speed breeding approach to introduce the hst1 (OsRR22) gene from the salt-tolerant Kaijin rice cultivar into the high-yielding Yukinko-mai background using MAS in less than 17 months [299]. Thus, speed breeding presents a viable way to quickly create stress-tolerant cultivars for food security.

Apart from multi-omics-assisted breeding techniques like MAS and metabolite-assisted selection, speed breeding can help minimize the expenses and land needed to produce a large number of cultivars. Producing distinct, uniform, and stable (DUS) data more quickly—a prerequisite for plant breeder rights—can hasten the early registration of novel cultivars [300]. Fast-track breeding and large-scale phenotyping of accessions for traits associated with salt tolerance can be facilitated and accelerated by integrating speed breeding with other technologies such as genome editing, phenomics, genomic-assisted breeding (GAB), and machine learning [292,301].

Research has shown that cool-season grain legumes, including peas, chickpeas, faba beans, lentils, and lupins, have a very precise flowering timing that is greatly influenced by adjusting light quality, especially the red-to-far-red ratio. These results are essential for accelerating single seed descent (SSD) in the breeding of legumes and provide useful applications in biotechnological tools for improving legume crops [302]. In another study, recombinant inbred lines with a salt tolerance emphasis were created in chickpeas utilizing SSD. Several QTLs linked to salinity tolerance were successfully discovered by researchers, offering important information for creating chickpea genotypes that are salt-smart [303]. Similar methods have been applied in studies on peanut breeding, where SSD, constant light, ideal temperature, and controlled environments were coupled in a greenhouse setting. Efforts in these areas have regularly shortened the generation times of full-season maturity peanut cultivars from 145 to 89 days. The novel speed breeding method has the potential to significantly accelerate the creation of peanut varieties, potentially reducing the conventional six- to seven-year breeding cycle from the initial cross to commercial release [304].

In lentils, speed breeding research is considered to be in its early stages. Mitache et al. (2024) investigated how lentil growth and development were affected by light intensity in fast-breeding settings [305]. They found that under minimal stress conditions, lentils reached optimal growth, flowered, and matured in 27 and 46 days, respectively. These findings strongly suggest the use of speed breeding to expedite the development of salt-tolerant lentil cultivars.

## 3. Improvement Strategies for Lentil Under Salt Stress

The development of future saline-smart lentil cultivars may involve various strategies. The following approaches should be prioritized (Figure 4).

### 3.1. Mini-Core Collections Development

The process of employing numerous genotypes to investigate the mechanism of lentil salt tolerance is expensive, time-consuming, and labor-intensive. By decreasing the size and increasing the diversity of the germplasm pool, it is possible to create a core library of salt tolerance that encompasses the full spectrum of genetic variability with minimal redundancy. This core set can serve as a starting point for optimizing genetic gains and more rapidly utilizing omics techniques. Utilizing phenotypic features from the core collection or trait-marker associations has proven to be a cost-effective, efficient, and labor-saving method for researching different response mechanisms [306].

### 3.2. Enhance Omics Pace

Using a variety of high-throughput approaches, it is possible to accelerate the improvement of lentils under salt stress while gaining a thorough understanding of various aspects of lentil plant biology. Several omics approaches, including single-cell omics, miRNAomics, and epigenomics, have not yet been used for lentils under salt stress. Consequently, it is recommended that the use of omics approaches be improved when dealing with salinity stress in lentils.

### 3.3. Integrated Omics

The rapid advancement of high-throughput data production methods has enabled extensive multi-omics-based systems biology research across all biological fields. Proteomics, metabolomics, transcriptomics, genomics, miRNAomics, epigenomics, phenomics, ionomics, machine learning, and speed breeding data can provide insights into the expression of genes, transcripts, metabolites, and proteins. However, these massive datasets can be comprehensively annotated, assimilated, and modeled using systematic multi-omics integration to generate profound and detailed insights. In lentils under salt stress, integrating omics data with innovative omics techniques can help bridge the gap between genome and phenome, thereby facilitating the selection of desirable phenotypes based on genetic contributions for breeding objectives.

### 3.4. Gene Validation

In order for any breeding or genetic program to succeed, one of the fundamental elements for marker-assisted and genomic selection is genetic validation. Genetic validation looks at whether the same gene or QTL tends to be significantly detected when grown in different years or locales and whether testing in diverse genetic backgrounds can still identify its effect [98]. Additionally, confirming previously identified genetic markers under salt stress is beneficial for future lentil improvement research, as it reduces time and effort under changing climate.

### 3.5. Expand Exploration from Lab to Field

A rigorous process ensures the development of viable, safe, and advantageous genetically modified cultivars when moving from laboratory research to field application aimed at enhancing lentil resistance to salt stress. By expanding research from the lab to the field, researchers can effectively introduce salt-tolerant lentil cultivars that improve resilience and agricultural output in saline regions.

### 3.6. Incorporate Genes and Understand Mechanisms from Other Legumes

Using genes from closely related legume species can help transfer desirable genes into lentils. Genes from species within the same family (*Fabaceae*) are typically used to ensure compatibility and functionality. Therefore, it is recommended that the identified genes and discovered mechanisms from other legume crops be applied to lentils.

## 4. Summary and Future Directions

The use of state-of-the-art biotechnological instruments, such as omics methods and contemporary breeding strategies, has proven to be an exceptional way to advance breeding methodologies and produce future cultivars that are saline-smart. To unravel the intricate mechanisms and associated genes in response to salt stress, a variety of techniques are employed, including genomics, transcriptomics, proteomics, metabolomics, miRNAomics, epigenomics, phenomics, ionomics, machine learning, and speed breeding (Figure 1). These fast and high-throughput approaches uncover the molecular, genetic, transcriptomic, proteomic, and metabolic elements that drive these responses, laying the groundwork for a deeper understanding of tolerance mechanisms. Genomics advances like GAB, MAS, QTL mapping, and GWAS have achieved some progress in this quest. The time has come for comparable developments in these areas, along with other omics techniques, to accelerate the development of saline-smart lentils. Recent advances in sc-omics have made it possible to combine different omics fields coherently and to consolidate them at the sc-level. The development of saline-smart lentils will become more feasible with the integration of machine learning and speeding breeding. To fill in the gaps in our understanding of lentil research, these tools provide a foundation for building a multidimensional database.

Still, some omics fields are in their infancy and need to be employed in lentil studies under salt stress. Integrated omics is highly recommended to identify key genes and deeply understand the pathways and mechanisms associated with salt stress in lentils. Despite the extensive collection and preservation of lentil germplasm worldwide, it is essential to develop mini-core collections related to salt stress. Validate the previously identified genes in lentils under salt stress and incorporate the genes and mechanisms discovered in other related legume crops. It is also recommended that research be conducted under field conditions rather than solely in laboratories, as lentils have shown altered results when tested under field conditions. To put it briefly, integrated omics techniques are essential for developing lentils that can adapt and endure in the face of constantly changing environmental and climatic conditions. This will help achieve the Sustainable Development Goals (SDGs) set forth by the United Nations and FAO, promoting sustainable agribusiness and enhancing quality and food security.

## Figures and Tables

**Figure 1 ijms-25-11360-f001:**
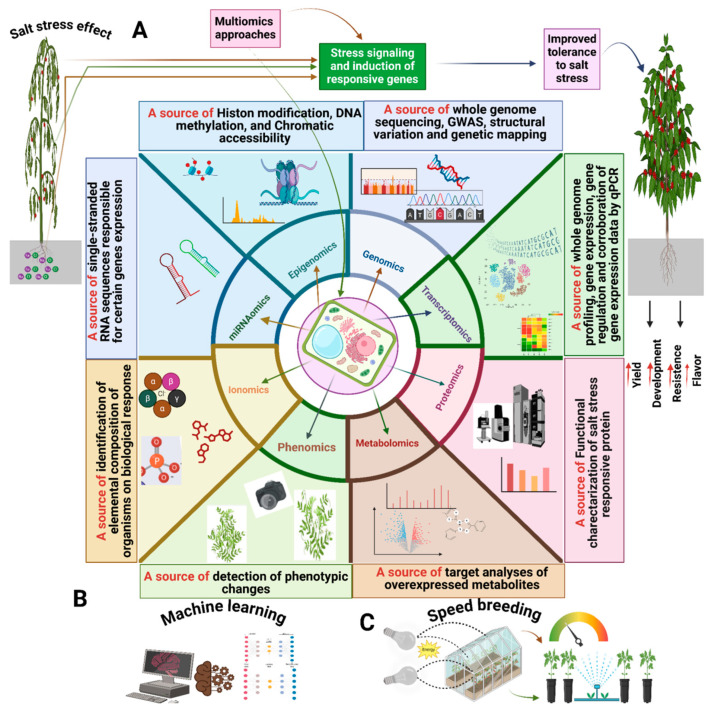
Salt stress affects lentil growth, but omics and other cutting-edge techniques help improve salt tolerance and develop saline-smart cultivars. (**A**) Salt stress from different sources in different forms affects lentil crops and reduces their growth, development, resistance, and flavor. Single omics or integrated omics, which involve combining two or more approaches, can be used in studies under salt stress alone or in combination with other stress conditions and plant tissues. These omics approach(es) produce comprehensive datasets that can be beneficial in developing saline-smart lentils. (**B**) Machine learning analyzes these datasets to determine how plants react to salt stress and to identify important components such as genes, metabolites, proteins, and markers. (**C**) Speed breeding facilitates the rapid introgression of salt-smart traits by accelerating breeding cycles. (**A**–**C**) Omics, machine learning, and speed breeding facilitate the development of saline-smart lentil cultivars. Created with BioRender.com.

**Figure 2 ijms-25-11360-f002:**
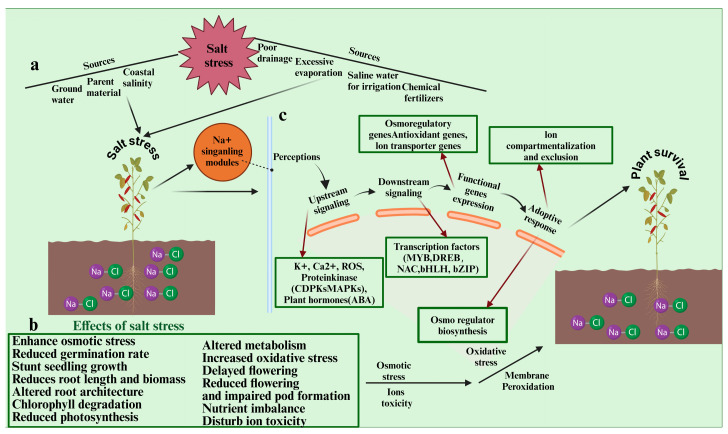
The graph represents the different natural and anthropogenic sources that cause salinity stress in lentils along with associated response mechanisms: (**a**) the different sources such as groundwater, parental material, coastal salinity, poor drainage, excessive evaporation, saline water for irrigation and chemical fertilizer causes a severe decline in lentil production, (**b**) the effect of salt stress on different growth-related traits, osmoregulators and osmoprotectants which directly involved in the yield and the quality of lentil, (**c**) the development of salt stress mechanisms to mitigate the salinity effect. Created with BioRender.com.

**Figure 3 ijms-25-11360-f003:**
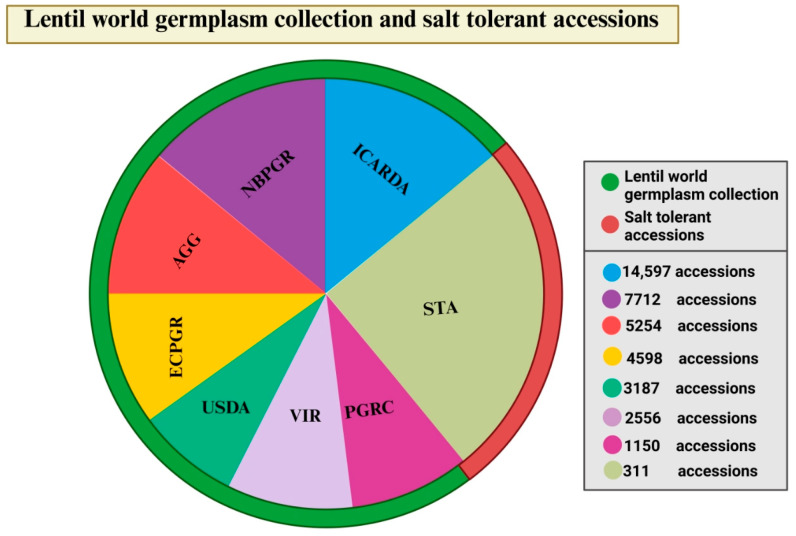
Lentil germplasm is collected and preserved around the world. ICARDA = International Center for Agricultural Research in the Dry Areas, NBPGR = National Bureau of Plant Genetic Resources, AGG = Australian Grains Genebank, ECPGR = European Cooperative Programme for Plant Genetic Resources, USDA = United States Department of Agriculture, VIR = the Vavilov Institute of Plant Genetic Resources, PGRC = Plant Gene Resources of Canada. STA = Salt-tolerant accessions. Created with BioRender.com.

**Figure 4 ijms-25-11360-f004:**
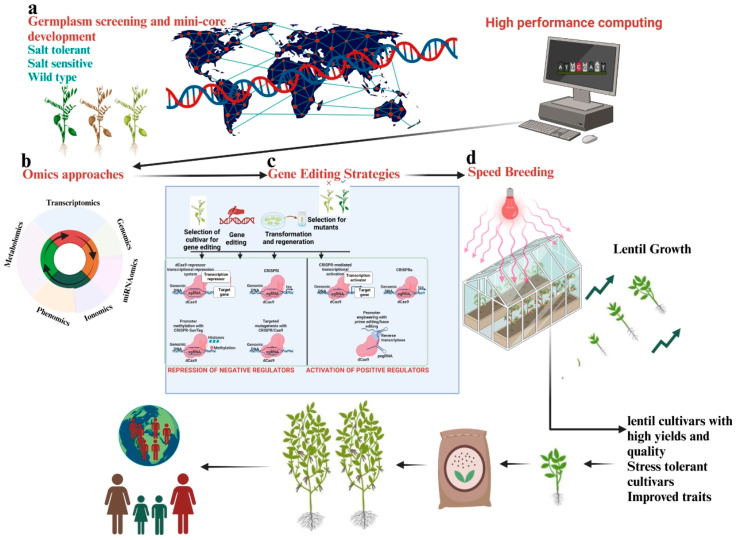
Omics-mediated options for creating salinity-smart cultivars include (**a**) germplasm collection and screening and mini-core collection development, (**b**) omics integration, (**c**) implementing advanced genetic engineering strategies, and (**d**) regular use of speed breeding. (**a**) Collecting and screening a wide range of germplasm from various gene banks and places around the world, including salinity-tolerant and -sensitive genotypes and their wild relatives. Recent improvements in high-performance computation, such as the use of multi-omics data, have tremendously aided crop genetic research. (**b**) Integrating various omics approaches can aid in the discovery of stress-related important actors, such as genes, proteins, metabolites, miRNAs, and metabolic pathways, which are critical in understanding salt stress responses and tolerance mechanisms. (**c**) Genetic engineering approaches, such as gene editing and transgenic breeding, have enormous potential for developing saline-smart future lentils to address world food security issues. Key actors identified by integrated omics can be genetically altered to control gene expression and the number of metabolites/proteins related to stress tolerance. The CRISPR system and introgression can be utilized to transmit beneficial genes isolated from wild relatives into cultivated varieties. (**d**) Speed breeding is the alteration of crop environmental conditions in order to accelerate the breeding cycle and advance to the next breeding generation as quickly as feasible [11,12,13]. This technology is useful for quickly producing multiple generations of modern saline-smart lentil genotypes. To summarize, integrating these instruments could empower plant scientists to produce saline-smart cultivars for farmers and growers, positioned to assist in feeding the ever-growing world population while also ensuring future food safety. Created with BioRender.com.

**Table 1 ijms-25-11360-t001:** List of the lentil salt-tolerant accessions.

S. No.	Accession Name/Number	Reference
1	Altın Toprak, Çağıl, Yerli Kırmızı and Fırat87	[52]
2	PDL-1 and PSL-9	[47]
3	PDL-1, PSL-9 and ILWL-09, ILWL-137, ILWL-96 and ILWL-428	[48]
4	42 salt-tolerant accessions	[50]
5	PDL-1	[53]
6	Firat87	[54]
7	KLS 218, Noori, L4076, HUL 57 and JL3	[55]
8	Bari Masur-4 and Bari Masur-5	[56]
9	Seyran	[58]
10	Giza 9	[59]
11	Castelluccio di Norcia, Pantelleria and Ustica	[60]
12	Jordan 1	[61]
13	BARI Lentil-7	[62]
14	Ustica and Pantelleria	[63]
15	Castelluccio di Norcia, Pantelleria and Ustica	[64]

**Table 2 ijms-25-11360-t002:** Epigenetic changes in legumes under salt stress.

Species	DNA Methylation/Histone Modification	Changes	Tissues, Organs, Genes/Location	Reference
Alfalfa	DNA methylation level at the promoter of MsMYB4	Decrease	Root	[236]
*Medicago truncatula*	Global DNA methylation	Increase	Root	[237]
*Medicago truncatula*	DNA methylation levels at CG, CHG, and CHH sites	Increase, decrease	374,944 differentially methylated sites in root	[235]
Pigeonpea	Genome-wide DNA methylation level	Decrease	Root and shoot	[238]
Soybean	DNA methylation level at the promoter of GmMYB84	Decrease	Leaf	[239]
Soybean	DNA demethylation at CG, CHG, and CHH sites	Increase	Glyma11g02400, Glyma16g27950, and Glyma20g30840	[240]
Soybean	Genome-wide DNA methylation levels at CG, CHG, and CHH sites	Decrease	Root	[241]
Alfalfa	H3K4me3 and H3K9ac	Increase	MsMYB4	[236]
Castor bean	H3K4me3 or H3K27me3	Increase, decrease	626 genes, including RSM1	[242]
Chickpea	H3K9ac	Increase	CaHDZ12	[232]
Soybean	H3K4me3	Increase	Glyma11g02400, Glyma08g41450, and Glyma20g30840	[240]
Soybean	H3K9me2	Decrease	Glyma11g02400, Glyma08g41450, and Glyma20g30840	[240]
Soybean	H3K9ac	Increase	Glyma08g41450 and Glyma20g30840	[240]
Soybean	H3K27me3	Increase, decrease	Increase in 5 expressed genes and decrease in 336 expressed genes	[243]

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
