# Peer review of "Omics-Driven Strategies for Developing Saline-Smart Lentils: A Comprehensive Review"

_ijms, 2024, doi:10.3390/ijms252111360_

Round 1
Reviewer 1 Report
Comments and Suggestions for Authors
The article with the title „ Omics-Driven Strategies for Developing Saline-Smart Lentils: A Critical Review” is very well written and comprehensive with many fields discussed regarding saline smart lentils.
The review aim is to discuss briefly all recent advances under climate change of lentil production under salinity stress.
Perhaps the title could be changed from a Critical review to A comprehensive review.
Some aspects are highlighted below:
The abstract is well summarized and well written.
The introduction is appropriate following the chosen subject for the review paper.
Figure 1 is very well done, congratulations.
Table 1 please remove the stress column because it is no change through species and it is already written in the table caption that is about salt stress.
All subchapters were treated accordingly very well done! Omics approaches were reached and described.
I especially like the following statement from the conclusion section: „Integrated omics techniques are essential for developing lentils that can adapt and endure in the face of constantly changing environmental and climatic conditions. This will help achieve the Sustainable Development Goals (SDGs) set forth by the United Nations and FAO, promoting sustainable agribusiness and enhancing quality and food security.”
Congratulation for the authors that they decided to explore such a novel and of high interest subject.
Reviewer 2 Report
Comments and Suggestions for Authors
The review manuscript by Fawad Ali et al. targets an interesting topic and could attract citations.
Positives:
The review outlines different aspects of salt-stress response study in lentils, which is promising.
Negatives:
The review does not fully deliver on the promised topics. Many parts are repetitive. The writing style is not very engaging. Portions of the manuscript seem to be stitched-together abstracts of the referenced manuscripts. The authors did not attempt any conclusive summary or data mining of the collected data.
Major issues
1) The review spans over 20 pages, yet the quantity is not always equivalent to quality. The review manuscript should engage the audience and provide a conclusive summary. It should not be repetitive, nor contain filler text that is irrelevant to the reviewed topic. I will list a few examples found in the manuscript to illustrate what should be improved in the entire review.
(a) Lines 541-542 present that legume crops are sensitive to salt stress. Information is missing references, and a similar text is already found in the introduction (lines 48-50) and in different iterations in many other parts of the manuscript (e.g., line 416).
(b) Text that is not relevant to the topic of the review should be limited. Lines 376-378 reference techniques that have not been used for salt stress experiments. There is no need to reference these studies, especially given the fact that there is no conclusive evidence that these studies provided more than descriptive characteristics. Lines 669-670 - Aphanomyces root rot reference - not related to the topic.
(c) Fillers with no real meaning for the reviewed topic should be removed. Line 405 - "transcriptomics is one of the most important omics approaches for gene expression analysis" - transcriptomics essentially covers all techniques that analyze gene transcripts. There are no other "omics" that would do that. Lines 52-67 - The paragraph does not contain a single reference. It is not needed in this form.
2) References in the review should be up-to-date. There are many instances where the authors comment on the "present state" using seriously outdated references. For example, annual production in Australia (information likely not needed) is backed up by references [4] Spies & Woodgate, 2005; and [5] Maher et al., 2003. This is not acceptable.
3) The review should provide a critical assessment of the collected information, not seemingly blindly copy-paste highlights and abstracts of the reviewed manuscripts. This is the most critical part and must be addressed in the potential revision. The absence of this process can be exemplified in two chapters:
(a) Proteomics: There is an extensive description of individual proteins found in different studies listing the number of "upregulated" and "downregulated" (this term should not be used for proteins or metabolites) proteins found in historic reports using outdated instrumentation with very limited proteome coverage. However, newer and likely more relevant studies that should be the focus of the review are only mentioned with the number of identified proteins.
(b) Chapter on world germplasm collection: The authors list the number of accessions found in collections around the world. However, I would be surprised if there would be zero overlap as indicated in the pie chart. Is it really the number of unique accessions? Given the topic of the review, the authors should focus on salt-tolerant/resistant accessions. What is the portion of these identified in the germplasm collections?
4) What novel information was found by collecting data for this review? The increasing power of AI algorithms makes classical reviews obsolete and the correct prompt will return up-to-date information from the scientific literature without a need for reading through a review. What is the added value that this manuscript can provide? The authors should try to synthesize the knowledge. One option could be a real assessment of different omics described in the manuscript. What is the overlap found in these different studies? Which pathways are supported by more than one approach or more than one research article? What are the key markers suitable for breeding and exploring further? What are the differences between lentils and the best plant models (Arabidopsis, rice)?
5) The figures are not bad, but devoid of meaningful information. Some depiction of molecular mechanisms found in salt stress response would be a welcome addition and elevate the manuscript above popular science.
Minor issues
1) The structure of the review does not follow a logical hierarchy. miRNAs are a part of RNA metabolism and should be included in transcriptomics or follow that chapter.
2) Many typos indicate a need for professional proofreading and text editing. There are also formulations that will require not only grammar/spell check. For instance, line 328 claims that genes expressed under salt stress were identified by in silico expression analysis. That seems fairly impossible. Either these genes are putative salt-stress response genes found by ortholog search (thus there is no information on the expression level), or the analysis was not done in silico.
3) Scientific rigor should be upheld in a review. The abbreviations must be consistent (e.g., HSP70 is referenced by different forms), the terms "expression" and "overexpression" should be used only for genes (proteins and metabolites have abundances), "plant hormone protein" is not a generally recognized term, MDA and H2O2 are not enzymes and don't have "activity", etc.
4) The review should contain information on the limitations of the study - what were the criteria used for selecting reviewed literature?
Comments on the Quality of English Language
Needs improvement, there are many typos and incorrect formulations.
Reviewer 3 Report
Comments and Suggestions for Authors
Dear authors, Your review provides much data for developing lentil cultivars resistant to salt stress and can be useful as a literature source for legume researchers. In my opinion, your review is not very suitable for the journal IJMS and would be better if will be published in the journal Plants or Agriculture.
I have some points with which you must agree.
line 247 change Medusa truncatula to Medicago truncatula
line 316-323 this text move to section 2.4 Metabolomics
line 357 while with a capital letter
section Transcriptomics
line 446-458 You have to mention the crop to which these investigations are concerned.
Proteomics
line 488 NDPKs explain
line 498 Medicago truncatula is the model legume
Ionomics
line 798-799 this line needs revision
Author Response
Please seed the attachment.

Round 2
Reviewer 2 Report
Comments and Suggestions for Authors
The authors have addressed some of my comments and made corresponding revisions to the manuscript. However, the provided examples were merely representative, and these localized changes have not significantly improved the overall quality of the manuscript. While I acknowledge their efforts, I believe that the revised manuscript still falls short in addressing several key issues.
Major Issues
Comment 3a: The review should offer a critical evaluation of the collected information, rather than simply copying and pasting highlights and abstracts from the reviewed manuscripts. This is a fundamental issue that must be addressed in any potential revision. The lack of a critical assessment can be observed in two specific chapters...
> The revised manuscript continues to present lengthy descriptions of proteins and genes identified in each study without providing a comprehensive summary that addresses the accumulated bias in the scientific literature. It is well-established that different studies often report conflicting results. Therefore, when listing these reports, the review should offer a summary and highlight the shared mechanisms supported by multiple sources.
> Germplasm Collection: The authors have not adequately addressed my question regarding whether the reported numbers represent unique germplasms or if there is some overlap that should be disclosed.
Comment 4: What novel information was found by collecting data for this review?
> While pointing towards new directions is certainly expected, the review will fall short of its potential if it does not present any intriguing novelties or interesting contrasts to established plant models.
Comment 5: Missing molecular mechanism graphical summary
I understand that the authors may be hesitant to address this comment due to its connection to the previous two issues. Please note that while I will not hinder the publication of the review if this last major issue remains unaddressed, I will not give my consent if the previous comments are ignored.
Minor Issues
Comment 2: The original abstract of the referenced manuscript explicitly states that "In silico expression analysis showed that four out of eight PDI genes (CPDI2, CaPDI6, CaPDI7 and CaPDI8) were expressed under salt stress." This clearly indicates that the authors utilized existing expression data for data mining. The review's reinterpretation of this information not only omits the original citation for the expression analysis but also incorrectly suggests that the stress was conducted in silico.
Comment 3: The manuscript's editing appears incomplete, as HSP70 is still listed as "heat shock 70 kDa."
Comments on the Quality of English Language
some editing is needed
Round 3
Reviewer 2 Report
Comments and Suggestions for Authors
Some of my issues with the manuscript have not been addressed in fullness, but I don't have any new suggestions that could help in improving this work.
Comments on the Quality of English Languagelegible